# Improved Gingival Margin Stability with a Digital Workflow in Esthetic Crown Lengthening: A Single-Center, Interventional, Non-Randomized, Open-Label Comparative Clinical Study on 622 Teeth

**DOI:** 10.3390/healthcare13243194

**Published:** 2025-12-05

**Authors:** Sorin Gheorghe Mihali, Dan Loloș, Andreea Raissa Hojda, Bogdan Antonio Loloș, Roxana Oancea

**Affiliations:** 1Department of Prosthodontics, Faculty of Dentistry, “Vasile Goldiș” Western University of Arad, 94 Revoluției Blvd., 310025 Arad, Romania; mihali.sorin@uvvg.ro; 2Faculty of Dental Medicine, “Victor Babeș” University of Medicine and Pharmacy Timișoara, Eftimie Murgu Square No. 2, 300041 Timișoara, Romania; 3Faculty of Medicine, “Victor Babeș” University of Medicine and Pharmacy Timișoara, Eftimie Murgu Square No. 2, 300041 Timișoara, Romania; andreea.hojda@student.umft.ro; 4Faculty of Medicine, “Vasile Goldiș” Western University of Arad, 94 Revolutiei Blvd., 310025 Arad, Romania; lolos.bogdan@student.uvvg.ro; 5Department of Preventive and Community Dentistry, Faculty of Dental Medicine, “Victor Babeș” University of Medicine and Pharmacy Timișoara, 300041 Timişoara, Romania; roancea@umft.ro

**Keywords:** esthetic crown lengthening, gingival margin stability, digital workflow, surgical guide, supracrestal tissue attachment, smile design

## Abstract

**Background/Objectives**: The stability of the gingival margin after crown lengthening is a key determinant of esthetic success in anterior rehabilitation. Digital workflows have been proposed to improve surgical precision, but their long-term impact on marginal stability remains insufficiently substantiated. **Methods**: A total of 87 patients (622 maxillary anterior and premolar teeth) who underwent esthetic crown lengthening were retrospectively evaluated. Patients were allocated to either a digitally guided workflow (G1, *n* = 62) or a freehand approach (G2, *n* = 25). Gingival margin stability was assessed using standardized intraoral photography and calibrated digital measurements at baseline, 1–2 months, and 12 months. Recurrence, periodontal parameters, complications, and patient satisfaction (VAS) were recorded. Mixed-effects models accounted for the clustering of teeth within patients. **Results**: Both workflows achieved predictable crown lengthening outcomes. At 12 months, median gingival margin recurrence was significantly lower in the guided group (G1: 0.14 mm [IQR 0.10–0.19]) compared with the freehand group (G2: 0.27 mm [IQR 0.20–0.34]) (*p* < 0.001). Secondary surgical revision was required in 1.6% of G1 patients versus 16.0% of G2 patients (*p* < 0.001). Patient satisfaction was high overall but greater in G1 (mean VAS 9.65 ± 0.52) than in G2 (8.96 ± 0.73). No major biological complications occurred. **Conclusions**: Digitally guided crown lengthening resulted in improved gingival margin stability and reduced the need for secondary correction compared with the freehand approach. Precise control of the bone crest position relative to the planned gingival margin is critical for long-term esthetic success.

## 1. Introduction

The harmonious relationship between the gingival margin, the dental crown, and the smile line is fundamental in anterior esthetics. Even minor discrepancies in gingival contour can disrupt smile symmetry and negatively impact patient perception and confidence [1,2]. In cases presenting excessive gingival display or altered passive eruption, crown lengthening may be indicated to restore proper tooth proportions and establish a stable supracrestal tissue attachment [3,4,5].

Conventional esthetic crown lengthening is commonly performed using a freehand approach guided by clinical measurements and soft tissue sounding [6,7]. While predictable when performed by experienced clinicians, this technique remains sensitive to operator variability. Small deviations in incision or osteotomy level may result in asymmetry or unintended marginal rebound, which is particularly evident in high smile line patients where esthetic tolerance is extremely narrow [2,8,9].

Digital technologies have expanded the diagnostic and surgical planning capabilities in periodontal and restorative therapy. Digital smile design and virtual wax up workflows allow the ideal gingival margin to be defined based on facial, dental, and esthetic parameters [10,11,12]. When translated into 3D printed surgical guides, this information can be used to direct both soft tissue incision and the corresponding osteoplasty, improving accuracy and reducing reliance on intraoperative estimation [13,14,15]. Recent clinical reports suggest that digitally guided esthetic crown lengthening may enhance predictability and improve control of marginal positioning [14,15,16,17,18]. However, controlled evidence directly comparing guided and freehand protocols remains limited, and long-term outcomes have not been consistently established [8,16].

A critical unresolved question is the long-term stability of the gingival margin after digitally guided crown lengthening. Soft tissue rebound following surgery is well documented and influenced by periodontal phenotype, flap thickness, bone crest remodeling, and postoperative plaque control [5,8,9]. Whether guided workflows reduce marginal relapse by ensuring more consistent alignment between the planned and executed osteotomy level has not been clearly established.

Patient centered outcomes and perceived esthetic improvement have been noted in several clinical reports on esthetic crown lengthening, although in most cases the assessment was descriptive or limited to the early postoperative phase, without the use of standardized long-term patient reported outcome measures [19,20]. Recent digitally guided protocols have reported improvements in surgical precision and perigivival stability, but patient centered outcomes remain inconsistently documented, particularly at follow up intervals beyond six months [13,14,15,16,17,18].

Furthermore, only a limited number of investigations have analyzed outcomes simultaneously at the tooth level and at the patient level, while accounting for the fact that multiple treated teeth may belong to the same individual, which is necessary to avoid overestimating statistical significance [21].

We hypothesized that a digitally guided workflow would result in lower gingival margin recurrence at 12 months compared with a conventional freehand approach, even after adjusting for plaque control and baseline characteristics.

The objective of this study was to compare gingival margin stability at twelve months between a digitally guided workflow and a conventional freehand approach for esthetic crown lengthening, using calibrated clinical and photographic measurements at both the patient level and the tooth level.

## 2. Materials and Methods

This retrospective comparative clinical study was conducted at the private clinic Dental Concept by Dr. Mihali in Timișoara, Romania, following a standardized prosthetic and periodontal workflow. This was a single-center, interventional, non-randomized, open-label, comparative study. The study adhered to the principles of the Declaration of Helsinki and received approval from the Ethics Committee of the “Victor Babeș” University of Medicine and Pharmacy in Timișoara (Approval Nr. 105/06.01.2020, revised 2025). Written informed consent was obtained from all patients or their legal guardians. All clinical records, photographs, and intraoral scan files were stored in a password-protected digital archive in compliance with GDPR requirements.

Consecutive patients who underwent esthetic crown lengthening between January 2021 and December 2023 were reviewed. Eligibility required the presence of excessive gingival display or insufficient clinical crown height in the maxillary anterior or premolar region, with stable periodontal conditions at baseline, defined as full-mouth plaque and bleeding scores ≤ 20%. Only cases with complete photographic documentation and a minimum clinical follow-up of 12 months were included. Patients were not considered if active periodontal inflammation was present, if the gingival enlargement had a systemic or pharmacologic etiology, or if ongoing orthodontic or restorative treatment could alter gingival margin levels during the observation period. Smoking habits were considered, and individuals who reported smoking more than 10 cigarettes per day were excluded to avoid the confounding effect on soft-tissue healing. Pregnancy, breastfeeding, uncontrolled bruxism, and untreated dental pathology in the treated region were additional exclusion factors. The screening and allocation process is illustrated in Figure 1.

Patients were allocated to groups based on the workflow applied at the time of treatment, and no randomization was performed. Treatment allocation was not randomized, and patients selected the workflow after being informed about both options. In both groups, an initial diagnostic wax-up and clinical mock-up were used to visualize the planned esthetic outcome. However, in the guided group (G1), the wax-up was digitized and merged with intraoral scans to produce a surgical guide that transferred both the intended gingival margin and the target position of the alveolar crest. In this workflow, the mock-up was used only for esthetic preview and patient communication, while the surgical guide served as the definitive intraoperative reference.

For 44 patients (29 in G1 and 15 in G2), long-term documentation was available from the clinic records. For these patients, we evaluated the initial clinical and photographic documentation, the 12-month recall, and all subsequent recall visits that were registered in their charts, up to a maximum follow-up of 10 years. The same evaluation criteria were used at each timepoint to assess changes in gingival margin position, soft-tissue contour, and overall esthetic appearance over time. This extended follow-up was used descriptively to observe the evolution of the soft tissues, and was not included in the comparative statistical analysis.

Representative steps in digital planning and guide fabrication are presented in Figure 1 and Figure 2. The full operative workflow for the guided group (G1) is illustrated in Figure 3, Figure 4 and Figure 5, highlighting the transfer of the planned gingival margin and bone crest position to the clinical setting.

In the digitally guided workflow (G1), the mock-up was used exclusively for esthetic preview and patient communication. The digitized wax-up was merged with intraoral scans, and the resulting surgical guide served as the definitive reference for both the gingival incision and the osteotomy level. In contrast, in the freehand workflow (G2), the mock-up functioned as the primary intraoperative reference. Gingival margins were marked directly according to the mock-up, and the extent of osteoplasty was determined through clinical inspection and transgingival bone sounding, without a digitally pre-defined osseous reference. The corresponding clinical workflow is illustrated in Figure 6 and Figure 7.

A threshold of 4 mm between the gingival margin and the alveolar crest was applied as the main clinical criterion. The supracrestal tissue attachment is generally around 3 mm, and maintaining this dimension is essential for long-term soft-tissue stability. When bone sounding indicated a distance below 4 mm, gingivectomy alone was considered sufficient because the biologic dimension was preserved. When the distance was greater than 4 mm, osteoplasty was required to prevent coronal rebound of the gingival margin during healing. The surgical protocol was consistent across cases. Local anesthesia was administered (Articaine 4% with epinephrine 1:100,000; Ubistesin™, 3M ESPE, Seefeld, Germany). A limited facial full-thickness flap was raised without vertical releasing incisions to preserve papillary integrity. Osteotomy and osteoplasty were performed to establish approximately 3 mm from the intended final gingival margin to the alveolar crest. Bone contouring was achieved using piezoelectric instrumentation (Piezotome^®^ Cube, Acteon, Bordeaux, France) in denser cortical areas or rotary burs (FG Diamond Burs, Komet Dental, Lemgo, Germany) when broader remodeling was required. Exposed root surfaces were smoothed where necessary. The flap was repositioned without coronal advancement and sutured using 5-0 monofilament PTFE (Cytoplast™, Osteogenics Biomedical, Lubbock, TX, USA) or 5-0 polypropylene (Prolene^®^, Ethicon, Raritan, NJ, USA). Simple interrupted sutures stabilized the margin, and vertical mattress sutures were used selectively to support papillary contour, particularly in thin periodontal phenotypes.

Postoperative care included 0.12% chlorhexidine rinses for 10–14 days and modified brushing in the operated area during the first week. Healing was evaluated at 1 and 6 months. Definitive restorative procedures were initiated only after gingival stabilization. Patient-reported esthetic satisfaction was assessed at the 12-month recall using a 0–10 Visual Analog Scale (VAS), where higher scores indicated greater satisfaction with the final smile outcome.

Gingival margin stability was evaluated on standardized intraoral photographs taken at baseline, 1–2 months, and one year postoperatively. All photographs were taken in a reproducible frontal position using the same camera-to-patient distance and lighting conditions. Digital measurements were performed in ImageJ version 1.53t (National Institutes of Health, Bethesda, MD, USA), calibrated using the clinical crown length of the maxillary right central incisor as the reference scale. The vertical distance from the incisal edge to the gingival zenith was measured at each timepoint. Measurements were repeated at both the one-month and six-month intervals by the same examiner, with the mean values used for analysis.

Long-term clinical documentation (5–10 years), when available, was reviewed descriptively to illustrate extended stability but was not included in the comparative statistical analysis.

Statistical analysis was performed using SPSS Statistics Version 27.0 (IBM Corp., Armonk, NY, USA) and R software version 4.2.2 (lme4 and lmerTest packages) for mixed-effects modeling. Normality of continuous variables was assessed using the Shapiro–Wilk test. Continuous outcomes at the patient level were compared using ANCOVA models, with treatment group as a fixed factor and baseline plaque scores and age as covariates. Categorical variables were compared using the Chi-square test or Fisher’s exact test, as appropriate. All analyses performed at the tooth level accounted for the hierarchical data structure in which multiple teeth were nested within individual patients. Therefore, mixed-effects ANCOVA models were applied for continuous outcomes (Zenith mm, Recurrence mm 12 m, ΔPPD), with patient included as a random intercept to control for intra-subject clustering. For binary outcomes (BOP and secondary surgical revision at 12 months), generalized linear mixed-effects models with a logit link were applied. Fixed effects included treatment group (G1 or G2), tooth type (incisor, canine, premolar), and Plaque Sextant 12m % as a covariate when relevant. Effect estimates are reported with 95 percent confidence intervals (CI). No a priori sample size calculation was performed because this was a retrospective analysis of all consecutive eligible patients treated during the study period. However, post hoc estimates based on the observed effect size for recurrence indicated that the study was sufficiently powered to detect clinically relevant between-group differences. The study was not prospectively registered because it was designed as a retrospective analysis of routine clinical care. Sensitivity analyses restricting the model to cases limited to the anterior sextant (11–21) yielded comparable results, confirming the robustness of the findings.

## 3. Results

A total of 102 patients were screened, of whom 87 fulfilled all eligibility criteria and were included in the final analysis (Figure 8). Of these, 62 were treated with the digitally guided workflow (G1) and 25 with the freehand approach (G2). Baseline demographic characteristics were comparable between groups, with no significant differences in age or sex distribution (Table 1).

Across all included patients, a total of 622 maxillary anterior and premolar teeth were evaluated. At the 1–2 months follow-up, both workflows demonstrated predictable crown lengthening outcomes, with the gingival margins positioned near the intended reference line derived from the preoperative esthetic design. At 12 months, a slight coronal shift in the gingival margin was observed in both groups. However, the magnitude of recurrence was lower in the guided workflow. Median recurrence at the tooth level was 0.14 mm (IQR 0.06–0.22) in G1 and 0.27 mm (IQR 0.17–0.39) in G2 (ANCOVA tooth-level: *p* < 0.001), representing a statistically and clinically relevant improvement in gingival margin stability in favor of the guided protocol. The difference remained consistent when analyzed at the patient level using median-per-patient aggregation (ANCOVA patient-level: *p* < 0.001) (Table 2, Figure 9).

Stratified analysis by extent of treatment (central incisors only/up to canines/up to premolars) demonstrated the same directional effect, with no significant interaction between group and treated extension (Figure 10).

Periodontal soft-tissue maturation was also favorable in both groups. Mean change in vestibular probing depth from 1–2 months to 12 months was −0.35 mm in G1 compared with −0.27 mm in G2. ANCOVA for ΔPPD, adjusted for baseline PPD: *p* < 0.001, indicating slightly greater remodeling and consolidation of the marginal tissues when the guided workflow was used. Bleeding on probing decreased substantially in both cohorts during the healing period, but remained lower at 12 months in G1 (5.2%) than in G2 (10.8%), a difference that reached statistical significance (*p* = 0.019) (Table 3, Figure 11).

Plaque levels at 12 months remained below 20% in all cases, in accordance with the maintenance criteria established in the protocol, and did not differ significantly between groups, confirming adherence to hygiene recommendations.

Minor postoperative tissue responses were observed in both groups and were generally limited to transient soft-tissue inflammation, short-lasting dentin hypersensitivity, and, in a small number of cases, mild marginal bleeding during the first days of healing. All events were self-limiting and resolved without clinical intervention. These findings were recorded in 6 patients in G1 (9.6%) and in 3 patients in G2 (12.0%), with no statistically significant difference between groups (*p* = 0.721). Importantly, no cases of papillary height loss, delayed wound healing, soft-tissue recession, or esthetic contour defects were observed in either group.

Secondary surgical revision during the 12-month follow-up were uncommon but were distributed unevenly between groups. Only one patient in the guided workflow group required a secondary surgical refinement (1.6%, 1/62), whereas four patients in the freehand group required secondary surgical revision (16.0%, 4/25) (*p* < 0.001, Chi-square). All secondary surgical revisions were performed to correct partial coronal rebound of the gingival margin in esthetically sensitive areas. No cases of papillary loss, open embrasures, or persistent soft-tissue asymmetry were present following corrective treatment. These outcomes indicate a lower risk of clinically perceptible relapse and a reduced need for revision surgery when the digitally guided protocol is applied (Table 4, Figure 12).

Correlation analysis demonstrated that higher plaque accumulation values, even within clinically acceptable ranges, were associated with slightly greater gingival relapse at 12 months, although plaque was not a significant covariate in the ANCOVA model (*p* = 0.25), independent of group allocation (Figure 13). Age showed a mild inverse association with recurrence (non-significant in ANCOVA, *p* = 0.67). Sex did not significantly influence outcomes.

The mean VAS score at 12 months was 9.68 (SD 0.62). When analyzed by groups, G1 showed a mean score of 9.65 (SD 0.52), while G2 presented a lower mean value of 8.96 (SD 0.73). In G1, 79% of patients reported VAS = 10, and the remaining 21% scored 9. In G2, 52% reported VAS = 10, 28% scored 9, and 20% scored 8. The lower scores in G2 were predominantly associated with minor postoperative complications (transient gingival inflammation, minor postoperative bleeding, or temporary dentin hypersensitivity), as well as isolated cases requiring secondary surgical revision due to soft-tissue zenith relapse.

Taken together, these findings indicate that both workflows provided predictable clinical outcomes for esthetic crown lengthening. The digitally guided approach resulted in greater stability of the gingival margin over the 12-month period and a lower need for secondary corrective procedures, while postoperative tissue response and periodontal parameters remained comparable between groups. This suggests that the guided protocol may improve the reproducibility of margin positioning without increasing clinical complexity or patient burden.

## 4. Discussion

The findings of this study indicate that the marginal soft-tissue stability following crown lengthening is primarily determined by the accuracy with which the bone crest is positioned relative to the planned restorative outcome. The digitally guided approach allowed the transfer of the prosthetic design directly to the surgical field, providing a reproducible reference for determining the target bone level. In contrast, the freehand technique relied on visual assessment and transgingival sounding, which carries an inherent risk of leaving the crest more coronally than intended. This incomplete re-establishment of the supracrestal tissue attachment is a known determinant of coronal soft tissue rebound during healing [22].

The guided workflow demonstrated greater stability of the marginal contour over time, reflecting the precision with which the final bone architecture was established. These outcomes align with contemporary evidence emphasizing that bone-driven planning, rather than soft-tissue–based estimation, governs long-term gingival positioning [23]. Digital control reduces operator variability; however, stable results remain dependent on correct osteotomy execution and consistent periodontal maintenance [24,25].

Patient-reported outcomes were consistent with these clinical observations. The higher VAS scores in the guided group (G1) reflected greater esthetic satisfaction and perceived symmetry of gingival contours. Lower VAS values in the freehand group (G2) corresponded to cases exhibiting marginal rebound or requiring refinement. This highlights that even small deviations in gingival zenith position are perceptible and clinically meaningful to patients [26].

Additional insight was obtained from the subgroup of patients with extended recall documentation (29 in G1 and 15 in G2), evaluated descriptively and not included in the comparative analysis. Within this group, 84% of freehand cases demonstrated soft-tissue remodeling requiring refinement, whereas guided cases generally maintained stable margins over time. The pattern observed was consistent: when periodontal bone sounding was ≥4 mm, gingivectomy alone provided stable outcomes; when sounding was <4 mm and gingivectomy was performed without corresponding osteoplasty, marginal rebound occurred consistently, indicating incomplete re-establishment of the supracrestal tissue attachment [27,28].

These observations reinforce that stability is not determined by flap design, instrument choice, or suturing technique, but by the precise relation between the bone crest and the planned prosthetic contours. Mock-up guidance may assist esthetic visualization, yet it does not provide an osseous reference and therefore cannot predictably prevent rebound when the bone crest remains too coronal [29]. Current periodontal-prosthetic workflows converge toward bone-anchored planning to ensure biologic width integrity and controlled marginal positioning [30].

The periodontal phenotype may influence susceptibility to soft tissue remodeling, particularly in thin scalloped biotypes. However, phenotype was not systematically assessed in the present cohort and should be incorporated in future prospective analyses [31].

Overall, the results indicate that digitally guided crown lengthening enhances the consistency of marginal stability and improves patient-perceived esthetic outcomes.

The freehand approach remains feasible, but demonstrates higher variability and increased likelihood of requiring refinement when the bone crest is not precisely positioned according to the restorative plan. The clinical implication is clear: stable esthetic crown lengthening depends on controlling the bone architecture, not on soft-tissue approximation.

This study presents several limitations that should be acknowledged. The design was non-randomized, and treatment allocation reflected the workflow available at the time of care. Although baseline characteristics were comparable, patients selected the workflow after being informed about both options, which may introduce selection bias. The analysis was based on a single-center cohort, which limits broader generalizability. Periodontal phenotype was not formally assessed, despite its known influence on soft-tissue stability. No a priori sample size calculation was performed because the study followed a retrospective design; however, post hoc estimates based on the observed effect size suggested adequate statistical power for detecting clinically relevant differences. The investigation was not prospectively registered, as it analyzed routine clinical care. Furthermore, although all included patients had at least twelve months of follow-up, not all cases had extended documentation beyond this interval. These aspects should be considered when interpreting the results, and future prospective multi-center studies with standardized phenotype evaluation and pre-specified sample size calculations are needed to validate these findings.

## 5. Conclusions

The digitally guided protocol provided greater accuracy in establishing the gingival margin and resulted in lower recurrence compared with the freehand approach. Both workflows achieved satisfactory esthetic and functional outcomes, but the guided technique demonstrated improved reproducibility and reduced need for secondary correction. Patient satisfaction remained high and was associated with stable soft tissue architecture and harmonious smile integration.

## Figures and Tables

**Figure 1 healthcare-13-03194-f001:**
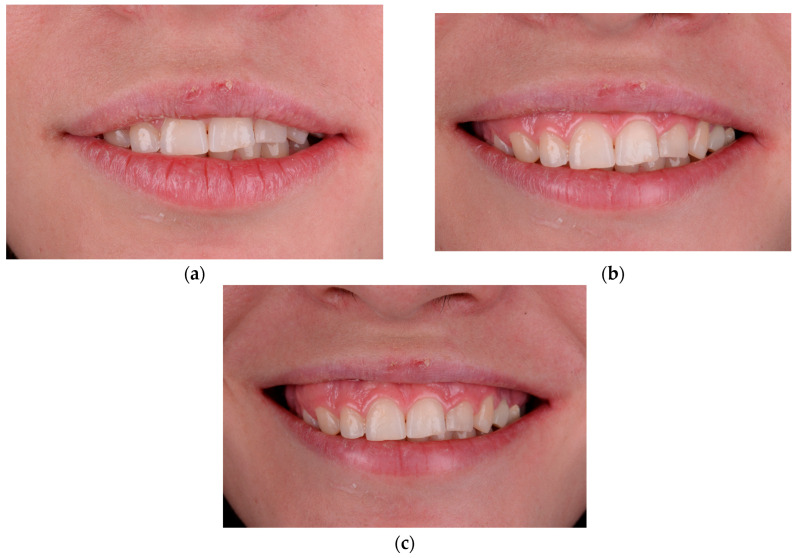
Baseline dento-facial records illustrating upper lip dynamics and initial gingival display: (**a**) lips-at-rest position, (**b**) social (posed) smile, and (**c**) full spontaneous smile.

**Figure 2 healthcare-13-03194-f002:**
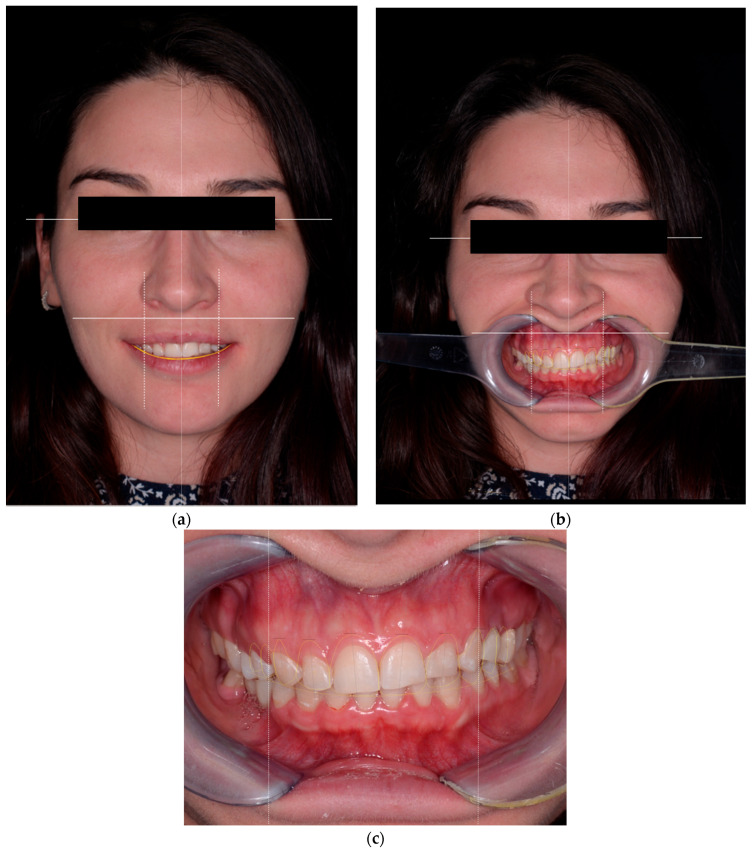
Initial dento-facial analysis used to determine tooth proportions and gingival display. (**a**) Frontal photograph at rest showing midline and facial reference axes. (**b**) Posed smile with lip elevation and smile line evaluation. (**c**) Intraoral frontal view illustrating gingival margin outline and clinical crown height relationships.

**Figure 3 healthcare-13-03194-f003:**
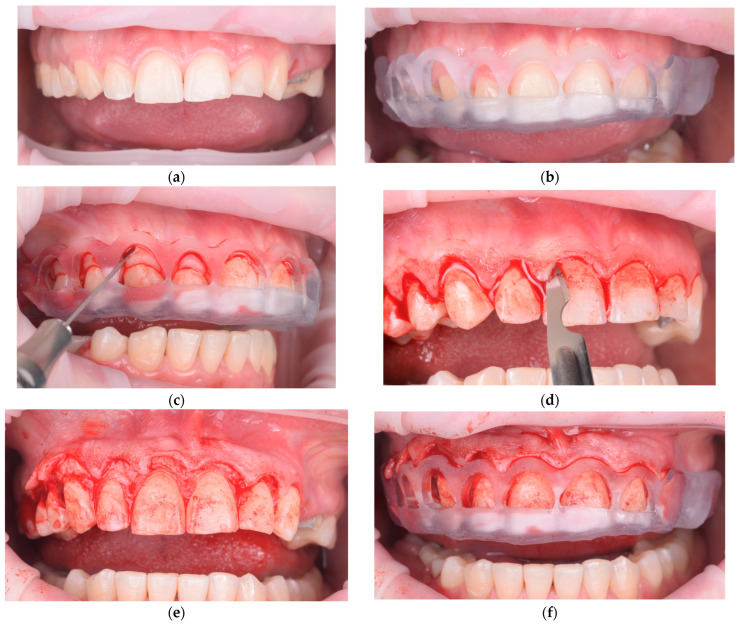
Guided surgical workflow for esthetic crown lengthening: (**a**) Preoperative frontal intraoral view showing excessive gingival display. (**b**) Digitally designed surgical guide positioned to transfer the planned gingival zeniths and bone crest level. (**c**) Marking of the new gingival margin through the guide for incision reference. (**d**) Gingivectomy performed according to the marked contour, maintaining papillary form. (**e**) Flap elevation and osteoplasty carried out to re-establish biologic width according to the planned reference. (**f**) Immediate postoperative view confirming alignment and symmetry of the definitive gingival margin.

**Figure 4 healthcare-13-03194-f004:**
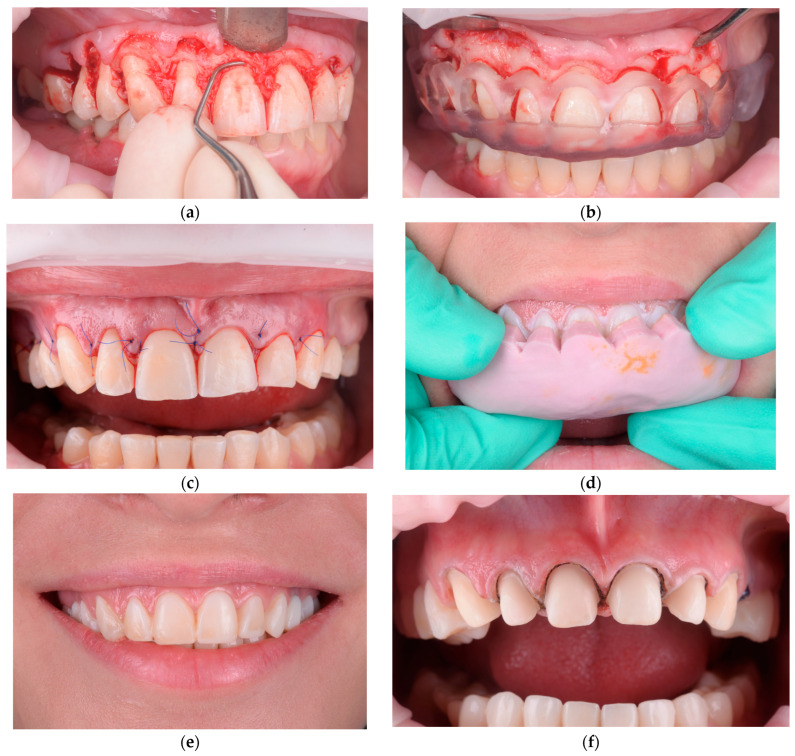
Freehand crown lengthening with mock-up validation: (**a**) Initial supra- and subgingival debridement (scaling and root planing) performed to reduce inflammation and clearly define the gingival margins before surgery. (**b**) Presentation of the newly established gingival architecture after soft-tissue contouring, highlighting the adjusted gingival zeniths. (**c**) Immediate postoperative intraoral view showing the harmonized gingival margin levels. (**d**) Insertion of the silicone mock-up intraorally to verify correspondence between the new gingival margin and the planned restorative contours. (**e**) Esthetic appearance after finishing and polishing the mock-up, illustrating the intended final tooth proportions and smile integration. (**f**) Tooth preparations performed prior to adhesive restoration, respecting enamel preservation and the newly stabilized gingival margins.

**Figure 5 healthcare-13-03194-f005:**
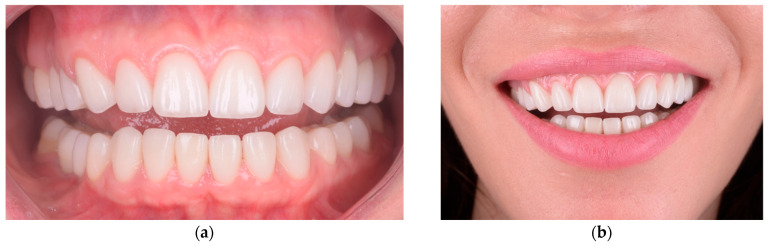
Final clinical outcome: (**a**) Intraoral frontal view after definitive restoration showing stable gingival margins, balanced crown proportions, and harmonious emergence profile. (**b**) Extraoral smile view demonstrating natural integration of the final esthetic result within the patient’s facial expression and smile dynamics.

**Figure 6 healthcare-13-03194-f006:**
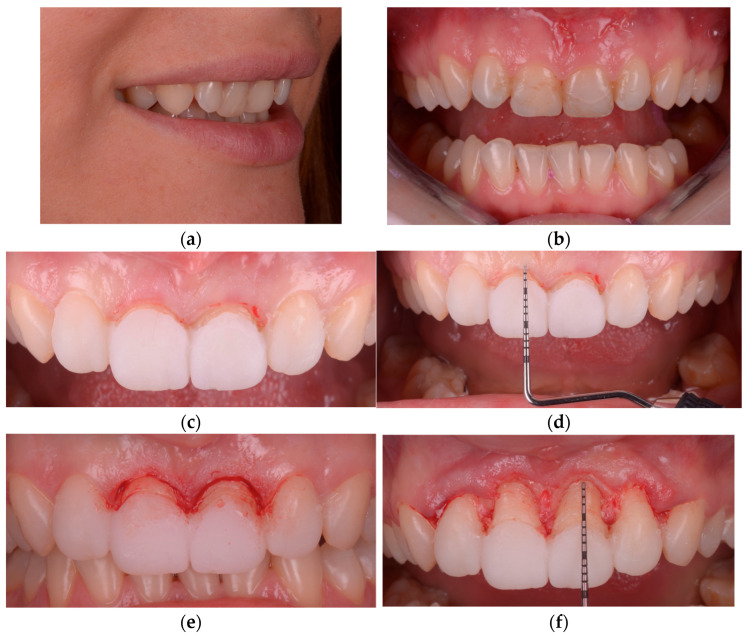
Freehand determination of gingival margins in G2: (**a**) Semi-profile smile view showing the palatal angulation of the maxillary central incisors and its influence on gingival display. (**b**) Initial intraoral frontal view illustrating reduced clinical crown height in the anterior maxilla. (**c**) Preliminary marking of the intended gingival zeniths using the mock-up as the esthetic reference. (**d**) Transgingival bone sounding performed to assess biologic width and determine the need for osteoplasty. (**e**) Freehand gingival contouring carried out according to the established reference line. (**f**) Verification of the final gingival margin relative to the underlying bone anatomy prior to restorative phases.

**Figure 7 healthcare-13-03194-f007:**
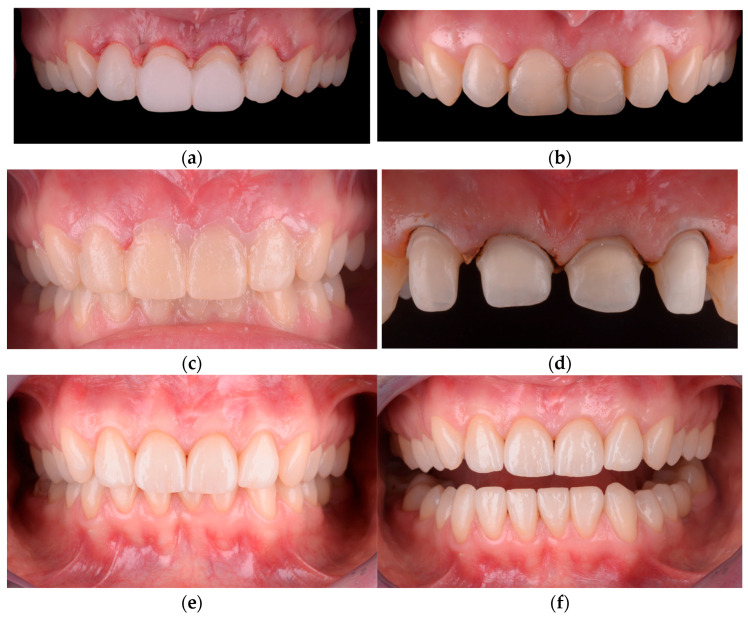
Soft-tissue healing and restorative sequence in G2: (**a**) Immediate postoperative intraoral view following freehand crown lengthening, showing the newly established gingival margins. (**b**) Intraoral view at 14 days demonstrating early soft-tissue healing and stabilization of the gingival contour. (**c**) Insertion of the mock-up after soft-tissue healing to verify tooth proportions, smile line, and gingival margin harmony before preparation. (**d**) Tooth preparations performed using a modified chamfer with a defined finishing shoulder, respecting the stabilized gingival zeniths. (**e**) Intraoral frontal view of the final restorations showing balanced crown proportions and stable marginal integrity. (**f**) Full-arch view of the final occlusion demonstrating functional and esthetic integration.

**Figure 8 healthcare-13-03194-f008:**
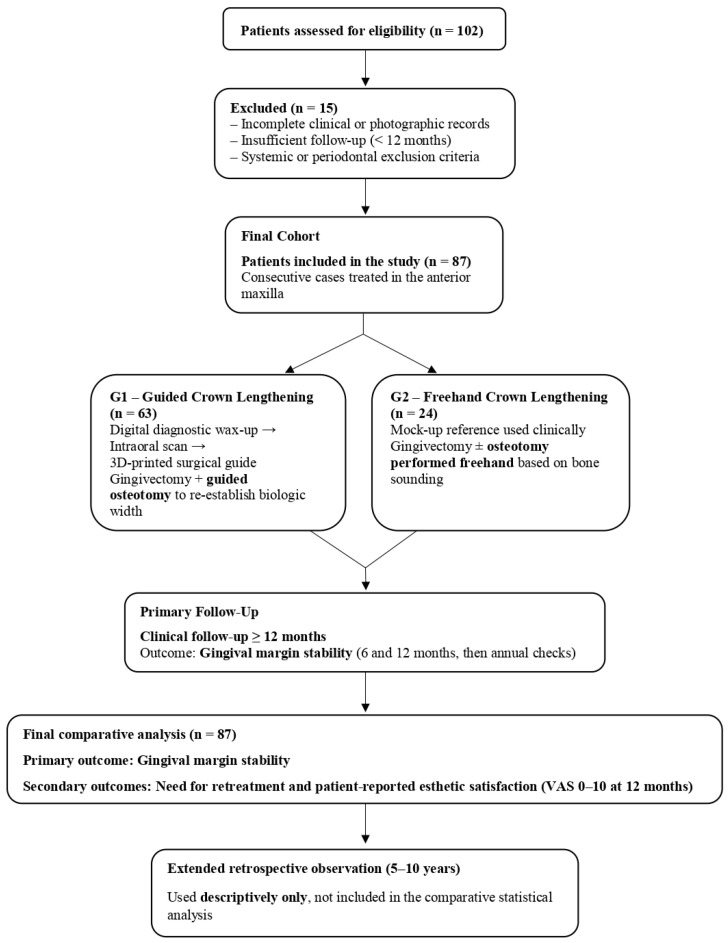
Flowchart of patient screening, group assignment, and follow-up period (*n* = 87).

**Figure 9 healthcare-13-03194-f009:**
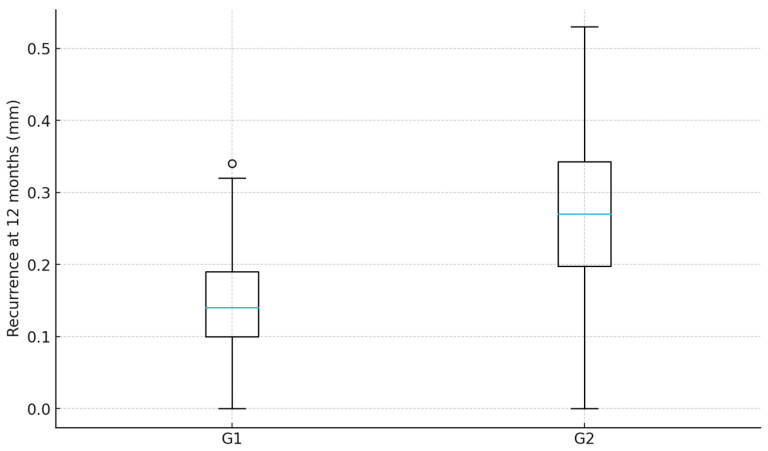
Recurrence at 12 months by group.

**Figure 10 healthcare-13-03194-f010:**
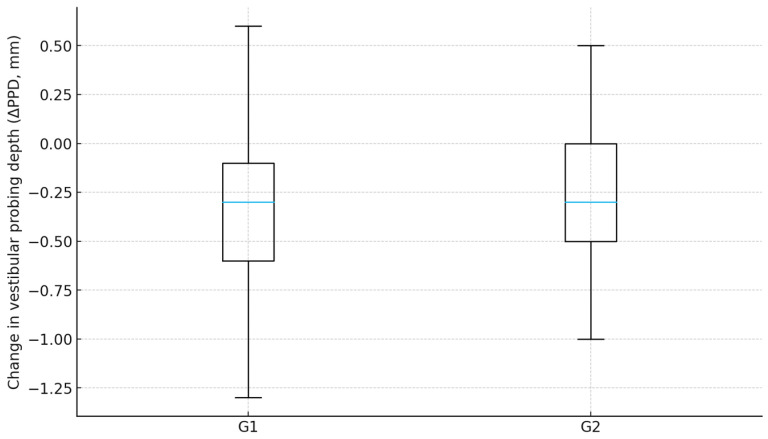
Change in probing depth by group.

**Figure 11 healthcare-13-03194-f011:**
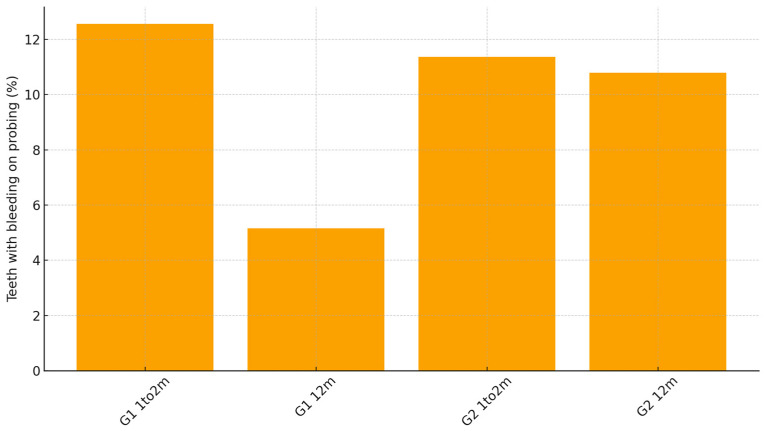
Bleeding on probing by group and time.

**Figure 12 healthcare-13-03194-f012:**
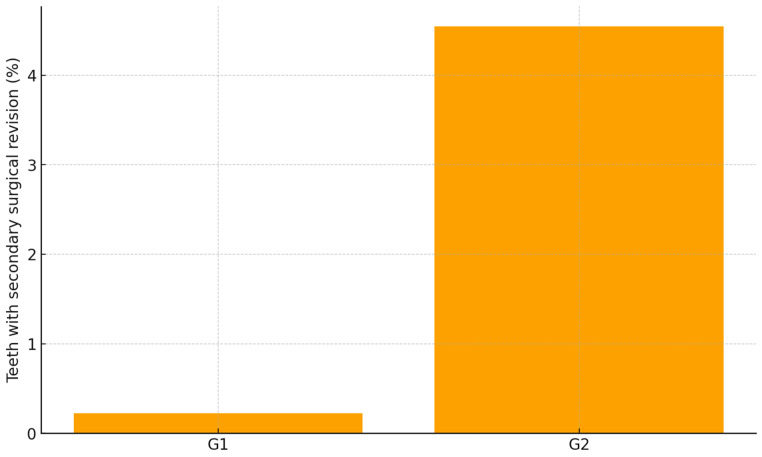
Secondary surgical revision within 12 months by group.

**Figure 13 healthcare-13-03194-f013:**
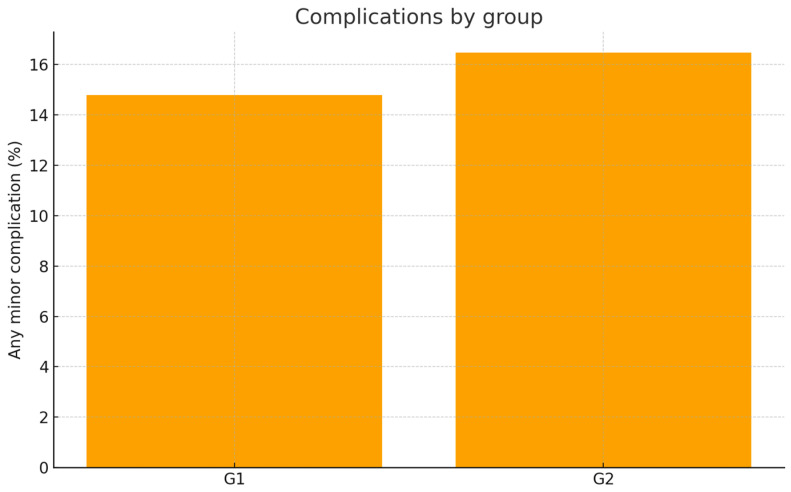
Minor complication frequency by group.

**Table 1 healthcare-13-03194-t001:** Baseline characteristics by group.

Group	N Patients	Female	Male	Age, Mean ± SD	Age, Median [IQR]
G1	62	37	25	31.11 ± 4.85	31.00 [28.00–34.00]
G2	25	17	8	31.16 ± 4.58	31.00 [27.00–33.00]

Sex distribution G1 vs. G2: *p* = 0.631. Age comparison (ANCOVA adjusted for sex): *p* = 0.84.

**Table 2 healthcare-13-03194-t002:** Primary outcome—coronal recurrence at 12 months.

Group	Per Tooth, Median [IQR] (mm)	Per Patient (Median-of-Teeth), Median [IQR] (mm)
G1	0.14 [0.10–0.19]	0.14 [0.12–0.16]
G2	0.27 [0.20–0.34]	0.27 [0.24–0.29]

Group effect tested by ANCOVA with Recurrence mm 12m as the dependent variable and Plaque at 12 months as covariate. Tooth-level analysis: *p* < 0.001. Patient-level analysis (median recurrence per patient): *p* < 0.001.

**Table 3 healthcare-13-03194-t003:** Periodontal parameters.

Group	ΔPPD (12m−1–2m), Mean ± SD (mm)	BOP 1–2m, % [95% CI]	BOP 12m, % [95% CI]
G1	−0.35 ± 0.31	12.6% [9.8–16.0] (*n* = 446)	5.2% [3.5–7.6] (*n* = 446)
G2	−0.27 ± 0.33	11.4% [7.5–16.9] (*n* = 176)	10.8% [7.0–16.2] (*n* = 176)

ΔPPD (G1 vs. G2) compared by ANCOVA with ΔPPD as the dependent variable and baseline PPD (PPD_1–2m_vest) as covariate: *p* < 0.001. BOP at 12 months, group comparison by Chi-square: *p* = 0.019.

**Table 4 healthcare-13-03194-t004:** Safety endpoints.

Endpoint	G1% [95% CI] (*n*)	G2% [95% CI] (*n*)	*p*-Value
Any complication	14.8% [11.8–18.4] (446)	16.5% [11.7–22.7] (176)	0.689
Secondary surgical revision ≤ 12m	0.22% [0.04–1.26] (446)	4.55% [2.32–8.71] (176)	0.000

Group comparisons for safety endpoints were performed using Chi-square or Fisher’s exact test, as appropriate.

## Data Availability

The data supporting the findings of this study are not publicly available due to patient privacy and ethical restrictions.

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
