# Peer review of "Improved Gingival Margin Stability with a Digital Workflow in Esthetic Crown Lengthening: A Single-Center, Interventional, Non-Randomized, Open-Label Comparative Clinical Study on 622 Teeth"

_healthcare, 2025, doi:10.3390/healthcare13243194_

Round 1

Reviewer 1 Report

Comments and Suggestions for Authors

The manuscript presents compelling evidence that a digitally guided workflow for esthetic crown lengthening leads to superior gingival margin stability at 12 months compared to a conventional freehand approach. Some minor modifications could improve the manuscript:

  1. Add aim of the study (and hypotheses ) at the end of the introduction.
  2. Methods; Provide a brief justification or reference for the specific 4 mm threshold used in bone sounding to decide between performing a gingivectomy alone or a gingivo-osteoplasty.

  3. Discussion Section: Explicitly state the study's limitations, including:

    • The Non-randomized design and the potential for selection bias.

    • The lack of a formal assessment of periodontal phenotype and its potential influence as a confounding factor.

    • Not all cases had long-term assessments
  4.  Ensure all References conform strictly to the journal's formatting guidelines.

  5.  Add more explanation regarding the role of the mock-up. State that in the digital workflow (G1), the surgical guide was the definitive reference for surgery, while in the freehand group (G2), the mock-up was the primary reference and so on.

Author Response

Dear Reviewer,
Thank you for your detailed comments and constructive feedback. Below I address each of your points in the order in which they were raised.

1. Comment: Add the aim of the study (and hypotheses) at the end of the introduction.
Response: In the original manuscript, the Introduction did not include a clearly stated aim. We have now added a final sentence that specifies the study objective. The new sentence reads: “The objective of this study was to compare gingival margin stability at twelve months between a digitally guided workflow and a conventional freehand approach for esthetic crown lengthening.” This clarifies the purpose of the investigation as requested.

2. Comment: Provide a justification for the 4 mm threshold used in bone sounding.
Response: The previous version only stated the clinical rule (“<4 mm gingivectomy, >4 mm osteoplasty”) without explaining the rationale. We expanded this section to clarify the biologic reasoning. The revised text explains that the supracrestal tissue attachment is approximately 3 mm and that maintaining this dimension is essential for soft-tissue stability. The paragraph now clarifies why gingivectomy is adequate below 4 mm and why osteoplasty is required above this value.

3. Comment: Add more explanation regarding the role of the mock-up and differences between workflows.
Response: The initial description of the workflows did not clearly differentiate how the mock-up was used in each group. This has now been clarified. The revised text states that in the digitally guided workflow, the mock-up served only as an esthetic preview and patient communication tool, while the surgical guide was the definitive intraoperative reference. In contrast, in the freehand group, the mock-up functioned as the primary intraoperative reference for marking margins and determining osteoplasty.

4. Comment: Explicitly state study limitations in the Discussion.
Response: The original Discussion section did not include a dedicated limitations paragraph. We added a new paragraph addressing the non-randomized design, the absence of periodontal phenotype assessment, and the variable duration of long-term documentation. This addition strengthens the transparency and interpretative balance of the section.

5. Comment: Ensure all references follow journal formatting.
Response: All references have been reviewed and adjusted to conform to the journal’s formatting requirements.

Thank you again for your helpful feedback, which contributed to improving the clarity and quality of the manuscript.

Best regards,
The Authors

Reviewer 2 Report

Comments and Suggestions for Authors

The manuscript presents a retrospective comparative study evaluating gingival margin stability after aesthetic crown lengthening performed either with a fully digital, guided workflow or with a traditional freehand approach. The authors include a relatively large sample of teeth and provide a minimum follow-up of 12 months, supplemented by long-term documentation for a subset of cases. The topic is clinically significant, especially as digital protocols continue to expand in periodontal and restorative practice. The manuscript is clearly written and easy to follow.

Overall, I found the article interesting and relevant for clinicians working in cosmetic dentistry and periodontology. The study compares two approaches that many practitioners consider in their daily practice, and the authors present their methods and results in a straightforward manner.

Key strengths include the number of teeth treated, consistent execution by a single experienced operator, and the use of appropriate statistical methods to treat grouped data. The results also appear consistent with what we observe in clinical practice, namely that guided approaches help standardize results and may reduce the risk of minor soft tissue recurrence.

However, because the study is retrospective and the differences between groups are sometimes small in magnitude, I believe that some clarification and a slightly more detailed discussion would strengthen the manuscript.

I believe the article can be accepted after minor revision.

Author Response

Dear Reviewer,
Thank you for your positive evaluation of our manuscript and for the constructive suggestions. Below we provide a point-by-point response addressing all aspects mentioned in your review.

Comment 1:
The manuscript is clearly written, clinically relevant, and the topic is important for practitioners in periodontal and restorative dentistry.
Response:
We thank the reviewer for the positive feedback and for highlighting the clinical relevance and clarity of the manuscript. We appreciate the acknowledgment of the study’s value for clinicians.

Comment 2:
The sample size is substantial, the treatments were performed by a single experienced operator, and the statistical approach is appropriate.
Response:
We appreciate the reviewer’s recognition of the methodological strengths of the study. No changes were required for this section, but we thank you for noting these elements.

Comment 3:
Because the study is retrospective and the differences between groups are sometimes small in magnitude, some clarification and a slightly more detailed discussion would strengthen the manuscript.
Response:
Following this recommendation, we reviewed the Discussion section and improved the clarity of several sentences to better highlight the clinical meaning of the findings. We also added a dedicated limitations paragraph, addressing the retrospective design, the absence of periodontal phenotype assessment, and the variability of long-term documentation. These changes strengthen the interpretative depth of the Discussion without altering its structure.

Comment 4:
The article can be accepted after minor revision.
Response:
We appreciate the reviewer’s overall positive judgment and support for acceptance after minor revision.

Best regards,
The Authors